# Palliative Sedation Therapy in Pediatrics: An Algorithm and Clinical Practice Update

**DOI:** 10.3390/children9121887

**Published:** 2022-12-01

**Authors:** Andrea Cuviello, Liza-Marie Johnson, Kyle J. Morgan, Doralina L. Anghelescu, Justin N. Baker

**Affiliations:** 1St. Jude Children’s Research Hospital, Memphis, TN 38105, USA; 2Department of Anesthesia, University of Tennessee Health Science Center, Memphis, TN 38105, USA

**Keywords:** palliative care, pain management, palliative sedation therapy, dexmedetomidine, propofol, symptom management, end of life

## Abstract

Palliative sedation therapy (PST) is an important clinical intervention for pediatric patients with refractory symptoms and suffering during the end-of-life (EOL) period. Variations in PST implementation including medication selection, limited literature regarding feasibility in various clinical settings, particularly non-intensive care units, and lack of education on evolving definitions and ideal practices may all contribute to the current underutilization of this valuable resource. We therefore offer a clinical algorithm for identifying appropriate patients for PST, ensuring all other modalities for symptom management have been considered and/or optimized, and present a guideline for PST implementation that can be adapted and individualized based on institutional experience and resource availability. Furthermore, through case-based clinical scenarios, we demonstrate how to incorporate this algorithm into EOL practice.

## 1. Introduction

Approximately 45,000 children will die each year in the United States [1]. Throughout the disease trajectory, and even when death becomes inevitable, our goal as healthcare providers is to cure when possible but moreover to relieve suffering. This goal to maximize comfort and minimize distressing symptoms is shared by patients and their families [2]. During the end-of-life (EOL) period, children can experience significant discomfort secondary to symptoms such as pain, dyspnea, fatigue and irritability: with the prevalence of these symptoms, particularly pain, reported to be nearly ubiquitous [3]. In some instances, traditional symptom management efforts are not enough to relieve a child of their suffering and it is during this time that palliative sedation therapy (PST) can be an asset.

PST is “the use of sedative medications to relieve intolerable and refractory distress by the reduction in patient consciousness”, a definition that spans mild to deep sedation but has often been associated with the latter [4,5,6]. Newer definitions for PST include terms such as “controlled sedation” and “proportional sedation” [7] and the American Academy of Hospice and Palliative Medicine (AAHPM) defined PST as “the intentional lowering of awareness towards, and including, unconsciousness for patients with severe and refractory symptoms” [8]. Nonetheless, PST can be a useful medical intervention to relieve suffering during the EOL period [9,10,11,12].

It is important to first distinguish that PST is not synonymous with physician-aid-in dying (PAD) or euthanasia, both of which often involve the use of medications to result in death [13,14]. Currently PAD is legal in only a handful of states (*n* = 10) within the USA [15] and often involves a rigorous screening process, a prescription from the physician, and finally, an active choice by the patient in self-administering a lethal dose of medication. It is also rarely implemented in pediatrics, with some European countries legalizing euthanasia [16]. In contrast, PST is the physician-controlled use of medications with the primary intention to achieve relief of refractory symptoms thereby relieving suffering. Medications are often titrated to effect with frequent re-assessment of symptoms and have been shown not to hasten death when used appropriately [17,18].

Opioids and benzodiazepines are among the most common medications utilized during the EOL period for the management of symptoms such as pain, dyspnea, and anxiety/agitation [19,20]. The side effect profile for both drug classes include reduction in patient consciousness and respiratory depression, the latter of which may cause death itself at some doses, although this is rarely encountered when appropriately trained clinicians utilize these drugs [21]. Nonetheless, this highlights a second important distinction when discussing PST, which is the principle of “double effect”. From an ethical perspective, the “double effect” principle states that a person is morally able to perform an act that has both good and bad effects if there is no other way to achieve the good effects, and if the following criteria are met: (1) the action itself is morally good, (2) the bad effect is not the means by which the good effect is achieved, (3) the motive must be the achievement of the good effect only and (4) the good effect is greater than or equal to importance to the bad effect [22]. Although the principle of “double effect” has often been suggested as a rationale for the use of opioids and benzodiazepines at the EOL, in reality, the use of these medications is very unlikely to cause death. Additionally, reductions in patient consciousness while using these medications to relieve symptoms is minimal. Furthermore, PST with these medications is becoming outdated with the introduction of alternative medications such as propofol and dexmedetomidine [23,24,25]. Ultimately, the principle of double effect can give rise to several ethical questions that are beyond the scope of this manuscript but may include “Is hastening death inherently bad?”, “Why is a non-treatment decision that may hasten death considered as good, while administering medication with the same effect considered bad?” and “Does morality rely on the act rather than the intention?”

This highlights the third important distinction when discussing PST, which is the intent with which medications are initiated. When a drug is initiated with the primary intent of reducing patient consciousness, this is referred to as “primary PST”. In contrast, when reduction in patient consciousness is a secondary effect of a drug whose primary use is for the management of a symptom (e.g., opioids for pain control), this is known as “secondary” sedation. It is important to make this distinction as it reflects on current clinical practice and helps define what providers classify as PST. For the purposes of this manuscript, the authors will be referring to PST in the context of primary PST.

### 1.1. Historical PST Practices

In a recent survey study completed by pediatric palliative care (PC) physicians and pain management specialists, significant variability was identified in the implementation of PST and which medications were chosen [26]. Among PC physicians, the most common class of drug used for PST implementation was benzodiazepines, followed by barbiturates [26]. In contrast, pain medicine specialists opted for opioids as a first line drug for PST initiation, followed by benzodiazepines [26]. This survey study highlighted additional variations in PST practice including lack of standardized protocols or guidelines for PST procedures, inconsistent involvement of ethics committees, and varying need for scope of physician orders (i.e., do not resuscitate orders) prior to PST initiation [26]. Overall, this study highlights the current variability in the clinical practice of PST and suggests targetable aspects for practice improvement.

### 1.2. New Directions in the Practice of PST

In an effort to improve the availability of PST, our group analyzed institutional PST practices over a 10-year period. We found that approximately 3% of all patient deaths and 12% of inpatient deaths utilized PST during the EOL [27]. In general, PST was implemented for refractory pain (33%), anxiety/agitation (17%), respiratory distress (13%) or a combination of these symptoms. Interestingly, we discovered a practice shift to include the use of dexmedetomidine in place of traditional sedatives like propofol [28]. Of 24 patients receiving PST, 83% (*n* = 20) achieved symptom control with dexmedetomidine and only one patient (4%) required a deeper level of sedation (with propofol) [27]. Among the 20 patients receiving dexmedetomidine infusions, approximately 50% (*n* = 11) required low-dose infusions (dosing range from 0.2 mcg/kg/h to 1 mcg/kg/h).

As a result of this institutional analysis, we redesigned our previously published algorithm for PST initiation and implementation. We present this algorithm (Figure 1) with the hope that it can serve as a model to be adapted and implemented at various institutions thus increasing the standardization of PST clinical practice. The remainder of this manuscript will utilize three case presentations to illustrate several key steps in the implementation of this algorithm, answering these questions: (1) Is the patient approaching the EOL? (2) Have multimodal pain interventions been optimized, and has the utilization of interventional pain management been considered? (3) How can low-dose dexmedetomidine infusions be employed as a “bridge” to a more sedation-based PST?

## 2. Case-Based Application of PST Algorithm

### 2.1. Case 1: “When PST Is NOT Indicated”

A 6-year-old male with relapsed, refractory acute megakaryoblastic leukemia and no further curative treatment options was admitted to the hospital for management of acute pain. At the time of admission, it appeared that the child was imminently dying, and he was experiencing intolerable suffering secondary to pain. The PC team was actively engaged with the patient and family and the primary goals of care included (1) comfort and (2) to maximize time spent at home. During the first 48 h of hospitalization, his analgesic regimen included a hydromorphone patient-controlled analgesia (PCA), which required rapid up-titration. As a result, the patient began to experience opioid-induced side effects including hallucinations, hyperalgesia, constipation and myoclonus, and he continued to have refractory pain.

Per our algorithm (Figure 1), interdisciplinary team assessment reflected that the patient was approaching EOL and experiencing intolerable suffering. Thus, a consult was placed for the pain management service. Family goals were confirmed (focus on comfort), therefore the next step was to optimize traditional symptom management interventions. A low-dose ketamine infusion was recommended, along with a subsequent weaning of opioids and aggressive management of opioid-induced constipation. The addition of low dose methadone was also recommended. The patient was ultimately able to achieve adequate pain control and be discharged home. He lived 2 more months at home with good symptom control through the EOL.

#### Take Home Points

This case exemplifies many important aspects when considering PST. First, it highlights the need for the optimization of multimodal analgesia prior to PST implementation. In this case, PST would have been inappropriate, primarily because pain management efforts had not been optimized. Second, it highlights the collaborative efforts that should be emphasized whenever available. In this case, the PC team was already involved, but pain medicine specialists were not. Pain medicine specialists, often anesthesiologists, can offer unique expertise related to symptom management throughout the disease trajectory and also during EOL [9]. As this case demonstrates, pain medicine specialists utilized advanced interventions such as a low-dose ketamine infusion to address factors such as opioid-induced hyperalgesia, central sensitization, and opioid tolerance. Interdisciplinary team collaboration can be helpful in optimizing symptom management. We also suggest including integrative medicine specialists, if available.

### 2.2. Case 2: “Thinking Outside the Box; An Example of Proportional Sedation Instead of PST to the Point of Unconsciousness”

A 22-year-old male with recurrent, widely metastatic Ewing sarcoma, no curative treatment options, and approaching the EOL, was suffering from diffuse pain secondary to tumor burden. Involved care teams included oncology, palliative care, integrative medicine, and pain medicine. Trialed pain medication regimens included ibuprofen, acetaminophen, cyclobenzaprine, lorazepam, opioids including fentanyl PCA, fentanyl lollipop and methadone, intermittent lidocaine infusions, low-dose ketamine infusion with eventual transition to PO ketamine, and multiple suprascapular nerve blocks (previously attempted to address shoulder pain secondary to primary tumor site). Despite these different modalities, the patient continued to experience diffuse refractory nociceptive and neuropathic pain.

Per the algorithm highlighted in Figure 1, traditional symptom management interventions have undergone a time-trial period and have not yielded acceptable symptom improvement. In response, an interdisciplinary team meeting with the patient and family members occurred to re-evaluate goals of care and begin discussions of interventions such as implanted pain catheter devices and PST. This meeting reaffirmed a goal of comfort, and that pain was the primary distressing and refractory symptom. Relief of pain was the primary objective shared between patient, family, and medical teams. As a result, an intrathecal catheter was suggested given the diffuse nature and nociceptive/neuropathic components of the patient’s pain and expected life expectancy of <3 months. An alternative option to PST, specifically low-dose dexmedetomidine infusion, to minimize sedation was also discussed. Due to patient-based reservations about foreign body implantation, a single intrathecal dose of morphine was trialed with no significant improvement. Consensus was reached to initiate low-dose dexmedetomidine (0.5 mcg/kg/h) which offered additional relief for pain, as well as anxiety, and did not alter consciousness at this dosing level. The patient died 3 days after initiation of dexmedetomidine, comfortable, and surrounded by loved ones.

#### Take Home Points

When pain control through traditional measures remains inadequate, or adverse effects become intolerable, peripheral nerve blocks or neuraxial (epidural or intrathecal) infusions can be used depending on the anatomic distribution of pain [30,31,32,33]. This case demonstrates an example of such a clinical scenario. While there is limited experience with the use of these modalities for pediatric patients at the EOL [31,32,34,35,36], central neuraxial blocks and continuous peripheral nerve blocks are increasingly used for pain control in adult EOL care [30,37,38,39]. One pediatric study demonstrated that continuous catheter-delivered pain blockade during the EOL contributes to analgesia and mitigates opioid requirements [30]. In this case, the patient continued to have pain despite all traditional and interventional pain management modalities, and per the algorithm, had proportional PST initiated with a low-dose dexmedetomidine infusion.

Dexmedetomidine has the potential to alleviate pain and contribute to adequate symptom control while simultaneously preserving a patients’ level of consciousness [40,41]. Through agonism of presynaptic alpha-2 adrenergic receptors, dexmedetomidine can lead to a reduction in the pain signaling pathway and has been shown to be both safe and effective in pediatrics [23]. More recently, a retrospective analysis showed a potential increasing role for dexmedetomidine at the EOL for symptom relief [27]. Once again, this case emphasizes the benefits of interdisciplinary team collaboration, specifically pain and palliative care services, to optimize symptom management for dying children.

### 2.3. Case 3: “Dexmedetomidine as a Bridge to PST to the Point of Unconsciousness”

A 19-year-old female was admitted for EOL care secondary to progressive, recurrent alveolar rhabdomyosarcoma. At the time of admission, she had significant pain that quickly became refractory to her current regimen which included hydromorphone PCA, methadone, topical lidocaine patch and lorazepam. Of note, the patient had a completed POST (DNaR order) several months before this admission.

Following our proposed PST algorithm, this patient was confirmed to be approaching EOL with refractory suffering from pain. Her goals were focused primarily on comfort, even if that meant dying in the hospital. Due to her disease status, anatomical distribution of her pain, and quick deterioration, interventions such as nerve blocks and implanted pain catheters were not considered. As a result, PST was discussed early during the hospitalization, and the goal of being symptom- and suffering-free was strongly emphasized by both patient and family. Low-dose dexmedetomidine was initiated with frequent reassessment and titration. Once dosing levels surpassed 1 mcg/kg/h, pain medicine and ethics teams were consulted for further evaluation of propofol-based PST. As the patient’s symptoms remained refractory despite increasing dexmedetomidine use, and distress to the patient was evident on exam and by communication with family, PST to the point of unconsciousness (with propofol) was initiated. The patient expired approximately 36 h after the initiation of propofol.

#### Take Home Points

This case presents PST to the point of unconsciousness. Reaching the end of our algorithm, through exhausting all traditional interventions, and being mindful of patient and family goals, PST was initially initiated with dexmedetomidine—which was subsequently titrated up to moderate to high dosing. Unfortunately, as symptoms continued to cause suffering, further progression to sedation to the point of unconsciousness with propofol was appropriate. Propofol is a drug with many advantages. Studies evaluating the propofol-opioid relationship demonstrate that: (1) propofol inhibits the metabolism of some opioids, thereby increasing their plasma concentration and (2) propofol and opioids interact synergistically, both enhancing pain control [42].

Throughout this admission for EOL care, high quality communication between patient, family and the medical team was critical. This can allow families to feel comfortable in advocating for additional symptom management needs, in this case proportional PST with eventual need for sedation to the point of unconsciousness. In addition, high quality communication and interdisciplinary team collaboration, including but not limited to primary medical team, pain medicine specialists, palliative care, nursing, social work, and ethics, can allow for PST to be initiated smoothly, based off the algorithm and guidelines, and ultimately minimize caregiver and provider distress. Opportunities for teams to debrief following a difficult EOL period can also help to identify areas for improvement and growth within an institution. Finally, bereavement support for families after the death of a child is a recommended psychosocial standard in pediatric oncology and should routinely be offered [43,44].

## 3. Conclusions

PST is a helpful medical intervention in alleviating refractory suffering for children at the EOL. Its growing role in the care of dying children should encourage increased research and education on PST and standardization of clinical practice. We describe an algorithm that can be used to decrease variability around PST practices and ideally lead to increased accessibility to this important tool.

## Figures and Tables

**Figure 1 children-09-01887-f001:**
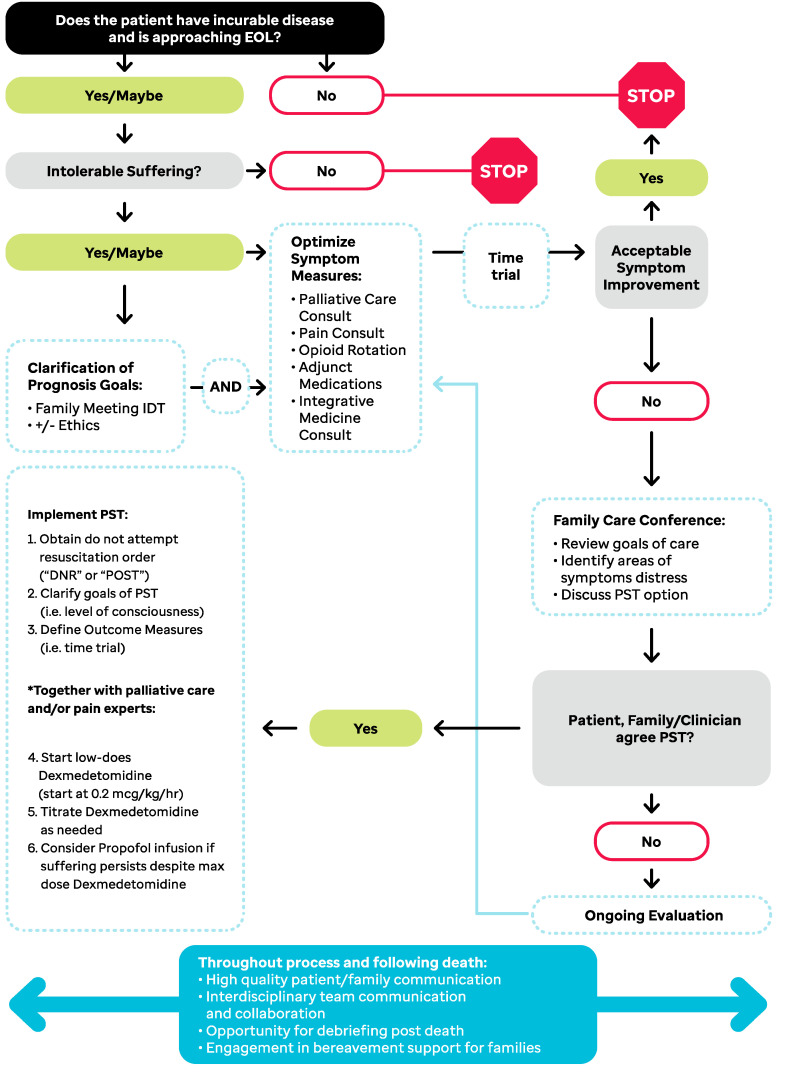
Algorithm for initiation of palliative sedation therapy (PST) * Maximum dosing for dexmedetomidine and starting dose recommendations for propofol not provided due to differences in physician prescribing patterns and influence of institutional policies on dosing. Previous author experience reports initiation of propofol at 30 mcg/kg/min [29].

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
