# Peer review of "Palliative Sedation Therapy in Pediatrics: An Algorithm and Clinical Practice Update"

_children, 2022, doi:10.3390/children9121887_

Round 1
Reviewer 1 Report
This paper reviews an important tool in hospice palliative care with which most providers outside the specialty have very little familiarity, and for which there currently exist no accepted guidelines, especially for pediatric patients. The inclusion of examples in which PST was ultimately not the right choice for the patient is also valuable and strengthens message that, in rare instances, it is truly the best choice to relieve suffering. My comments/suggestions are as follows:
1) The citation included in the first sentence (re: the number of pediatric deaths in the US annually) does not appear to contain this information. It's a great paper, but doesn't speak to the epidemiology of childhood deaths. I would suggest using a different reference to support this data. One suggestion would be the following, which estimates 55,000 childhood deaths annually in the US:
Institute of Medicine (US) Committee on Palliative and End-of-Life Care for Children and Their Families; Field MJ, Behrman RE, editors. When Children Die: Improving Palliative and End-of-Life Care for Children and Their Families. Washington (DC): National Academies Press (US); 2003. CHAPTER 2, PATTERNS OF CHILDHOOD DEATH IN AMERICA. Available from: https://www.ncbi.nlm.nih.gov/books/NBK220806/
2) Line 35: correct "relive" to "relieve."
3) Line 70: suggest rephrasing "define what providers are constituting as PST." The providers are not constituting, the providers are defining what constitutes. Consider "define what providers classify as PST."
4) Line 79: would specify ethics committees.
5) Lines 88-89: Was this practice shift recent? Gradual over the 10 years reviewed? What do the authors posit contributed to this shift?
6) Line 201, "continued to remain" is redundant.
7) Line 228: Should be "its," not "it's."
8) Figure 1:
-"Optimize Symptom Measures" could also include titration of existing medications.
-"Implement PST" defines low-dose dexmedetomidine with instructions to "titrate as needed," but does not suggest the threshold of "max dose" dexmedetomidine at which providers should consider adding propofol. Similarly, it would be useful to included a starting rate for propofol.
9) While it may be beyond the scope of this article, I'm curious as to whether the authors have an education resources they use to provide information to involved staff and patients/families who are considering PST?
Author Response
This paper reviews an important tool in hospice palliative care with which most providers outside the specialty have very little familiarity, and for which there currently exist no accepted guidelines, especially for pediatric patients. The inclusion of examples in which PST was ultimately not the right choice for the patient is also valuable and strengthens message that, in rare instances, it is truly the best choice to relieve suffering. My comments/suggestions are as follows:
1) The citation included in the first sentence (re: the number of pediatric deaths in the US annually) does not appear to contain this information. It's a great paper, but doesn't speak to the epidemiology of childhood deaths. I would suggest using a different reference to support this data. One suggestion would be the following, which estimates 55,000 childhood deaths annually in the US:
Institute of Medicine (US) Committee on Palliative and End-of-Life Care for Children and Their Families; Field MJ, Behrman RE, editors. When Children Die: Improving Palliative and End-of-Life Care for Children and Their Families. Washington (DC): National Academies Press (US); 2003. CHAPTER 2, PATTERNS OF CHILDHOOD DEATH IN AMERICA. Available from: https://www.ncbi.nlm.nih.gov/books/NBK220806/
Thank you for catching this oversight. This reference has been updated.
2) Line 35: correct "relive" to "relieve."
This has been corrected.
3) Line 70: suggest rephrasing "define what providers are constituting as PST." The providers are not constituting, the providers are defining what constitutes. Consider "define what providers classify as PST."
Thank you for this comment. The suggested rephrasing has been incorporated.
4) Line 79: would specify ethics committees.
Thank you. This has been adjusted accordingly.
5) Lines 88-89: Was this practice shift recent? Gradual over the 10 years reviewed? What do the authors posit contributed to this shift?
Thank you for this comment. Unfortunately, due to the survey study design, we were unable to assess the timing for this shift or variability in clinical practice.
6) Line 201, "continued to remain" is redundant.
Thank you. This has been adjusted to “remained”.
7) Line 228: Should be "its," not "it's."
Thank you. This has been corrected.
8) Figure 1:
-"Optimize Symptom Measures" could also include titration of existing medications.
-"Implement PST" defines low-dose dexmedetomidine with instructions to "titrate as needed," but does not suggest the threshold of "max dose" dexmedetomidine at which providers should consider adding propofol. Similarly, it would be useful to included a starting rate for propofol.
Thank you for this comment. With the goal of minimizing clutter on a visual aid for palliative sedation implementation the authors felts that the word “optimize” was synonymous with “titrate” and opted to remove this to avoid clutter/redundancy. In regard to the dosing of dexmedetomidine, we intentionally left out maximum dose ranges as this could be influenced by a) comfort level of the managing physician (our personal experience ranges from 0.5-2 mcg/kg/hr and even occasionally up to 4-5 mcg/kg/hr between our palliative care physician team), b) institutional policies which may limit maximum dosing levels especially if performed outside of an ICU setting and c) the clinical picture, as some patients may progress to requiring propofol. Propofol can be added as an additional layer to dexmedetomidine or replace the dexmedetomidine to achieve symptom management. This starting dose is again a) anesthesia physician dependent and b) may reflect institutional policies. A footnote to this point has been added to Figure 1.
“*Maximum dosing for dexmedetomidine and starting dose recommendations for propofol not provided due to differences in physician prescribing patterns and influence of institutional policies on dosing. Previous author experience reports initiation of propofol at 30mcg/kg/min (44).”
9) While it may be beyond the scope of this article, I'm curious as to whether the authors have an education resources they use to provide information to involved staff and patients/families who are considering PST?
Thank you for this comment. We currently do not have educational resources available to provide families and staff with regarding palliative sedation therapy. We are working to update our institutional policy for palliative sedation implementation to reflect the algorithm included in this manuscript. The authors agree that developing educational resources for patients/families/staff would be quite valuable.
Reviewer 2 Report
This paper describes an algorithm for the initiation of palliative sedation therapy in pediatrics. Additionally, cases are described to illustrate when palliative sedation therapy would be considered beneficial or contra-indicated. I agree that the field needs guidelines on palliative sedation for pediatrics, as this is currently lacking. However, I have some issues regarding parts of the introduction, the classification of the paper as a review and the definition of palliative sedation therapy used. I would therefor recommend a major revision before considering this paper for publication in Children.
I have the following concerns/questions:
For me, the description of palliative sedation therapy as a tool, rather than a medical treatment or medical intervention seems a bit odd.
If I am not mistaken, at least in Belgium, it is not PAD but rather euthanasia that is considered legal (in specific circumstances) for minors.
For me, the important difference between PAD or euthanasia and palliative sedation is not that, when it is titrated correctly, the medication does not hasten death. The most important difference is that the intention for medication administration is not to hasten death. This might imply that hastening death is often due to inadequacy of the physician, when I can imagine that it is sometimes the goal of the physician to do so (whether or not this is legal is another matter entirely).
The introduction might benefit from a critical note on the doctrine of double effect. Is hastening death inherently bad? Why is a non-treatment decision that might hasten death considered as good, while administering medication with that same effect is considered bad? Does morality rely on the act rather than the intention?
I would not consider this paper to be classified as a review.
In adults, palliative sedation is defined as reducing consciousness to treat refractory symptoms at the end of life. As such, I wonder whether palliative sedation without reducing consciousness is still palliative sedation and not simply adequate pain and symptom management? Are both considered as ‘official’ types of palliative sedation or is this something that is introduced by the current paper? If so, I would like this to be more clear.
I agree that a family care conference is preferable, but I wonder if this is always an option. I can imagine cases where this is not so. Is palliative sedation than not considered to be an option?
Author Response
This paper describes an algorithm for the initiation of palliative sedation therapy in pediatrics. Additionally, cases are described to illustrate when palliative sedation therapy would be considered beneficial or contra-indicated. I agree that the field needs guidelines on palliative sedation for pediatrics, as this is currently lacking. However, I have some issues regarding parts of the introduction, the classification of the paper as a review and the definition of palliative sedation therapy used. I would therefor recommend a major revision before considering this paper for publication in Children.
I have the following concerns/questions:
For me, the description of palliative sedation therapy as a tool, rather than a medical treatment or medical intervention seems a bit odd.
Thank you for this comment. We have adjusted the language throughout the manuscript to use the term “medical intervention” instead of “tool".
If I am not mistaken, at least in Belgium, it is not PAD but rather euthanasia that is considered legal (in specific circumstances) for minors.
Thank you for this comment. You are correct and this has been reflected in the manuscript which now reads: “It is also rarely implemented in pediatrics, with some European countries legalizing euthanasia [12].”
For me, the important difference between PAD or euthanasia and palliative sedation is not that, when it is titrated correctly, the medication does not hasten death. The most important difference is that the intention for medication administration is not to hasten death. This might imply that hastening death is often due to inadequacy of the physician, when I can imagine that it is sometimes the goal of the physician to do so (whether or not this is legal is another matter entirely).
Thank you for this comment. We have made an adjustment to the wording to clarify this point. The manuscript now reads: “In contrast, PST is the physician-controlled use of medications with the primary intention to achieve relief of refractory symptoms thereby relieving suffering. Medications are often titrated to effect with frequent re-assessment of symptoms and have been shown not to hasten death when used appropriately [13, 14].”
The introduction might benefit from a critical note on the doctrine of double effect. Is hastening death inherently bad? Why is a non-treatment decision that might hasten death considered as good, while administering medication with that same effect is considered bad? Does morality rely on the act rather than the intention?
Thank you for this comment. Lines 55-67 currently describe the doctrine of double effect and how it relates to medications used for palliative sedation therapy. The authors appreciate and agree with the importance of the ethical questions you mention, however answering them directly is challenging, controversial and outside of our goals for this paper. We have chosen to highlight these points within the manuscript with the below text in the hopes that they may spark intellectual conversation and though among potential readers.
“Ultimately, the principle of double effect can give rise to several ethical questions that are beyond the scope of this manuscript but may include “Is hastening death inherently bad?”, “Why is a non-treatment decision that may hasten death considered as good, while administering medication with the same effect considered bad?” and “Does morality rely on the act rather than the intention?’”
I would not consider this paper to be classified as a review.
Thank you for this comment. The authors wholeheartedly agree and are working with the editorial committee to confirm the best classification for this manuscript. Initially this paper was designed as more of a clinical practice update or guideline for pediatric providers. The authors felt that this would be most useful as there is currently little literature on pediatric palliative sedation and the recent works that have been published stem from our author prior work products.
In adults, palliative sedation is defined as reducing consciousness to treat refractory symptoms at the end of life. As such, I wonder whether palliative sedation without reducing consciousness is still palliative sedation and not simply adequate pain and symptom management? Are both considered as ‘official’ types of palliative sedation or is this something that is introduced by the current paper? If so, I would like this to be more clear.
Thank you for this comment. We have clarified this in the introduction which now reads: “PST is “the use of sedative medications to relieve intolerable and refractory distress by the reduction in patient consciousness”, a definition that spans mild to deep sedation but has often been associated with the latter (4-6). Newer definitions for PST include terms such as “controlled sedation” and “proportional sedation”(7) and the American Academy of Hospice and Palliative Medicine (AAHPM) defined PST as “the intentional lowering of awareness towards, and including, unconsciousness for patients with severe and refractory symptoms”(8). Nonetheless PST can be a useful medical intervention to relieve suffering during the EOL period (9-12).”
We have also added a clarification statement (underlined text) to the section discussing primary PST and secondary sedation (lines 81-85) which now reads: “When a drug is initiated with the primary intent of reducing patient consciousness, this is referred to as “primary PST.” In contrast, when reduction in patient consciousness is a secondary effect of a drug whose primary use is for the management of a symptom (e.g., opioids for pain control), this is known as “secondary” sedation. It is important to make this distinction as it reflects on current clinical practice and helps define what providers classify as PST. For the purposes of this manuscript, the authors will be referring to PST in the context of primary PST.”
I agree that a family care conference is preferable, but I wonder if this is always an option. I can imagine cases where this is not so. Is palliative sedation than not considered to be an option?
Thank you for this comment. We agree that in some cases a family conference is not feasible given the acuity of the clinical situation. Under these circumstances, discussion with patients and caregivers regarding palliative sedation as an option for symptom management would still occur, just on more intimate level, rather than an organized interdisciplinary team meeting. As this is a recommended, not required step, we try to complete a family care conference whenever possible.
Round 2
Reviewer 2 Report
Thank you for your clear answers to my comment. I feel that the changes made definitely make the paper more clear. Congratulations.